# RISK: A Framework for GUI Agents in E-commerce Risk Management

## Abstract

E-commerce risk management requires aggregating diverse, deeply embedded web data through multi-step, stateful interactions, which traditional scraping methods and most existing Graphical User Interface (GUI) agents cannot handle. These agents are typically limited to single-step tasks and lack the ability to manage dynamic, interactive content critical for effective risk assessment. To address this challenge, we introduce RISK, a novel framework designed to build and deploy GUI agents for this domain. RISK integrates three components: (1) RISK-Data, a dataset of 8,492 single-step and 2,386 multi-step interaction trajectories, collected through a high-fidelity browser framework and a meticulous data curation process; (2) RISK-Bench, a benchmark with 802 single-step and 320 multi-step trajectories across three difficulty levels for standardized evaluation; and (3) RISK-R1, a R1-style reinforcement fine-tuning framework considering four aspects: (i) Output Format: Updated format reward to enhance output syntactic correctness and task comprehension, (ii) Single-step Level: Stepwise accuracy reward to provide granular feedback during early training stages, (iii) Multi-step Level: Process reweight to emphasize critical later steps in interaction sequences, and (iv) Task Level: Level reweight to focus on tasks of varying difficulty. Experiments show that RISK-R1 outperforms existing baselines, achieving a 6.8% improvement in offline single-step and an 8.8% improvement in offline multi-step. Moreover, it attains a top task success rate of 70.5% in online evaluation. RISK provides a scalable, domain-specific solution for automating complex web interactions, advancing the state of the art in e-commerce risk management.

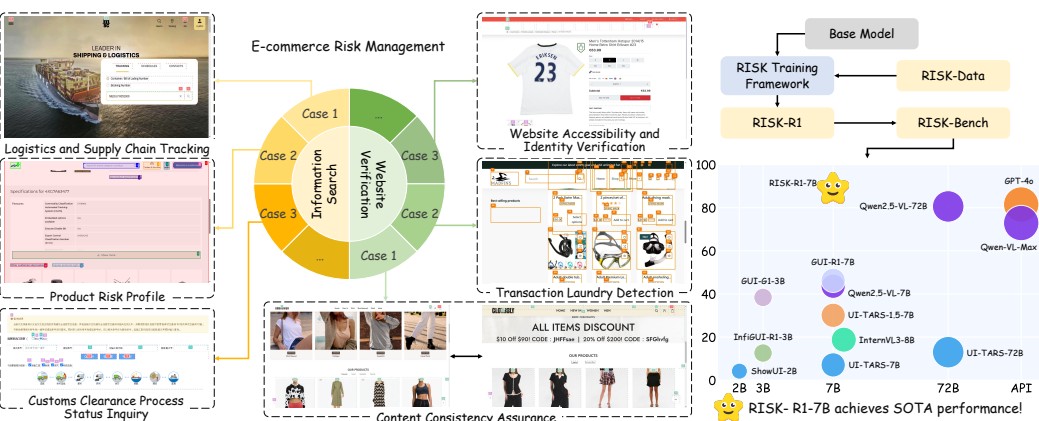

Figure 1: RISK framework for GUI agents in e-commerce risk management. Left: Task composition for GUI agents in e-commerce risk management, including information search and website verification tasks. Right: RISK framework, which consists of three key components: RISK-Data, RISK-Bench, and RISK-R1. RISK-R1 achieves SOTA performance in this domain.

## 1 Introduction

In e-commerce transaction scenarios, stringent compliance and risk control mechanisms are essential to mitigate operational, regulatory, and reputational risks. Decision-making in this context requires the aggregation of heterogeneous information from multiple external sources, many of which exist

as unstructured or semi-structured data on the public web. While broad web search can identify relevant sources, truly actionable intelligence often resides deep within specific websites—sometimes on dynamically loaded subpages, behind interactive elements, or embedded within complex document object models (DOM). This sophisticated web navigation and data extraction process costs significant manual effort and domain expertise, making it a prime candidate for automation through intelligent agents (Yoran et al., 2024; Ning et al., 2025).

Traditional scraping APIs or static crawlers fail to retrieve such deeply embedded content, as they lack the ability to engage in stateful, event-driven interactions (Petrova et al., 2025). Recently, GUI agents (Gu et al., 2025; Lin et al., 2025; Qin et al., 2025; Liu et al., 2025) powered by multimodal large language models (MLLMs) (Bai et al., 2025; Zhu et al., 2025; Anthropic, 2024; Hurst et al., 2024) have shown promise in automating web navigation and interaction tasks. These agents can interpret visual and textual cues on a webpage, plan the action sequence, and execute interactions to achieve specific goals. Current mainstream GUI agents focus on data-driven training paradigms and have increasingly adopted the reinforcement fine-tuning (RFT) paradigm (Luo et al., 2025; Zhou et al., 2025; Tang et al., 2025; Yuan et al., 2025). Through carefully designed reward functions, RFT could guide the learning process of MLLMs and enhance their grounding capabilities in GUI tasks.

Despite the rapid progress of MLLM-driven agents, most existing Web GUI agents in both academia and industry remain limited to executing single-step operations reliably. This single-step paradigm, while functional for simple actions, fails to support end-to-end e-commerce risk management tasks in realistic web environments, where multi-step reasoning, dynamic content handling, and complex interaction sequences are required. Moreover, the lack of domain-specific datasets and benchmarks further impedes the development of GUI agents tailored for this area.

To harness the full potential of GUI agents in this domain, we propose a novel Web UI agent framework, called RISK, which comprises three key components: (1) RISK-Data. RISK-Data is collected using the Browser Use framework (Müller & Žunič, 2024), which is a framework that integrates advanced context management, optimized prompt templates for both page screenshots and HTML DOM structures, and precise low-level interaction capabilities. We aim to systematically distill and embed the framework's advanced knowledge into the data, thereby improving the success rate of multi-step, real-world web workflows. After a meticulous curation process, RISK-Data contains 8,492 single-step and 2,386 multi-step interaction trajectories on various task scenarios, shown in Figure 1. (2) RISK-Bench. RISK-Bench is collected for evaluating the performance of GUI agents in e-commerce risk management. It consists of 802 single-step and 320 multi-step trajectories, which are graded into three difficulty levels: easy, moderate, and difficult. (3) RISK-R1. RISK-R1 is an RFT framework based on Group Relative Policy Optimization (GRPO) (Shao et al., 2024). We design a framework-driven reward function and optimization objective to effectively guide the learning process of GUI agents and enable a seamless transition from training to deployment. Specifically, there are four key aspects: (i) Output Format: Updated format reward that enhances the syntactic correctness of the model's output and task understanding, (ii) Single-step Level: Stepwise accuracy reward that measures action accuracy considering both action completeness and training process, (iii) Multi-step Level: Process reweight that emphasizes the step stage in the interaction process, and (iv) Task Level: Level reweight that focuses on different difficulty levels of tasks.

Experiments on RISK-Bench demonstrate that our approach achieves substantial gains over existing baselines in e-commerce risk management tasks. In offline evaluation, RISK-R1-7B improves single-step performance by 6.8% and multi-step performance by 8.8%. In online evaluation, it attains a top task success rate of 70.5%. Comprehensive analysis further validates the synergistic contributions of each component in RISK-R1. Our contributions are summarized as follows:

- We introduce the RISK framework, which integrates domain-specific data collection, benchmarking, and reinforcement fine-tuning for GUI agents in e-commerce risk management.
- We develop RISK-Data, a high-quality dataset with 8,492 single-step and 2,386 multi-step interaction trajectories, and RISK-Bench, a benchmark with 802 single-step and 320 multi-step trajectories for evaluating GUI agents in this domain.
- We propose RISK-R1, a novel RFT approach based on GRPO, with a comprehensive reward function and optimization objective to enhance the learning process of GUI agents and facilitate deployment in real-world applications.
- Extensive experiments demonstrate that RISK-R1 outperforms existing baselines, achieving SOTA results in both offline and online evaluations on e-commerce risk management tasks.

## 2 RELATED WORK

### 2.1 GUI AGENTS

GUI agents are intelligent systems that can understand and interact with graphical user interfaces through various actions (e.g., click, type), to accomplish automated execution of complex GUI tasks (Sun et al., 2024; Zhang et al., 2024; Tang et al., 2025; Hu et al., 2025). GUI agents can be broadly categorized into three types: (1) Expert knowledge-driven workflow. These agents construct a workflow consisting of two main components (Li et al., 2024; Wang et al., 2025; Xie et al., 2025; Zhang et al., 2025; Jiang et al., 2025): planner and actioner, where planner decomposes high-level tasks into sub-tasks and generates corresponding action sequences, and actioner is responsible for providing an accurate element localization (e.g., bounding box, DOM tree index). However, these agents heavily rely on expert knowledge to design the workflow and cause error accumulation in long-horizon tasks. (2) Data-driven training. These agents are MLLMs trained on GUI understanding and interaction datasets through supervised fine-tuning (SFT) (Wu et al., 2024; Xu et al., 2024; Qin et al., 2025; Lin et al., 2025) or reinforcement fine-tuning (RFT) (Luo et al., 2025; Zhou et al., 2025; Tang et al., 2025; Liu et al., 2025). Rather than decomposing tasks into sub-tasks, they can end-to-end generate actions based on the current GUI state and task instructions. However, it is challenging to deploy these agents in real-world applications due to the poor generalization ability on unseen webpages and the expensive, high-quality data collection. (3) GUI agent framework. Recently, frameworks such as WebVoyager (He et al., 2024), OpenManus (Liang et al., 2025) and Browser Use (Müller & Žunič, 2024) garner significant attention in the GUI agent community for several reasons: (1) Customized GUI agents by integrating various LLMs and tools, (2) Enhanced context management and more rigorous handling of tool I/O parameters, (3) Capability to interact with real-world webpages and collect complete trajectories for further model training. Given these advantages, a practical approach to domain-specific GUI agents is to use these frameworks to collect domain-specific data, fine-tune MLLMs (supervised or reinforcement), and redeploy the trained models within the frameworks to further enhance their performance for real-world applications.

### 2.2 RFT IN GUI AGENTS

Following the release of DeepSeek-R1 (Guo et al., 2025), RFT with rule-based rewards have been widely adopted (Zhang & Zuo, 2025; Feng et al., 2025; Huang et al., 2025). As RFT is anticipated to tackle the problem of poor generalization in SFT, it is also introduced in GUI tasks (Yuan et al., 2025; Liu et al., 2025). Currently, the mainstream methods focus on designing reward functions to guide the learning of grounding capability. For instance, GUI-R1 (Luo et al., 2025) takes the prediction point within the bounding box of the target element as a successful action and assigns a binary reward accordingly. GUI-G1 (Zhou et al., 2025) further considers the relative size of the grounding box and designs a more fine-grained reward function. GUI-G2 (Tang et al., 2025) proposes a Gaussian continuous reward mechanism for a flexible evaluation of grounding accuracy. However, while interacting with real-world webpages, GUI agent frameworks (Müller & Žunič, 2024) do not use (x,y) coordinates for element selection but employ the element index in the DOM tree combined with various tools. Therefore, the gap between the GUI model training and deployment settings makes the existing reward functions inapplicable, and then the model cannot be employed in GUI agent frameworks flexibly for real-world applications. To address this issue, we propose an RFT framework, named RISK-R1, to train GUI agents for e-commerce risk management. RISK-R1 designs a comprehensive reward function and optimization objective to effectively guide the learning process of GUI agents and enable a seamless transition from training to deployment.

## 3 DATASET COLLECTION

Currently, open-source datasets in GUI agents (Kapoor et al., 2024; Chai et al., 2024; Li et al., 2025) are general and lack domain-specific tasks. To address this gap, we propose a comprehensive pipeline for collecting and curating a domain-specific dataset tailored for GUI agents in e-commerce risk management, called RISK-Data. Moreover, to quantitatively assess the performance of GUI agents in this specialized domain, we introduce a novel benchmark named RISK-Bench.

### 3.1 TASK DESIGN

In practical applications in the e-commerce risk management domain, GUI agents are required with the following capabilities: (1) **Information Search**: GUI agents should be able to efficiently nav-

igate through various webpages and interfaces to locate specific information, such as transaction details, user profiles, and historical data. This involves understanding the structure of webpages, recognizing relevant elements, and executing appropriate actions to retrieve the needed information. (2) **Website Verification**: GUI agents must be capable of verifying the authenticity and security of websites. This includes checking for secure connections (e.g., URL redirection), validating certificates, and identifying potential phishing or fraudulent sites. The ability to discern trustworthy sources is crucial in mitigating risks associated with online transactions. Developed based on these capabilities, our task composition is shown in Figure 1 and detailed in Appendix A.2.

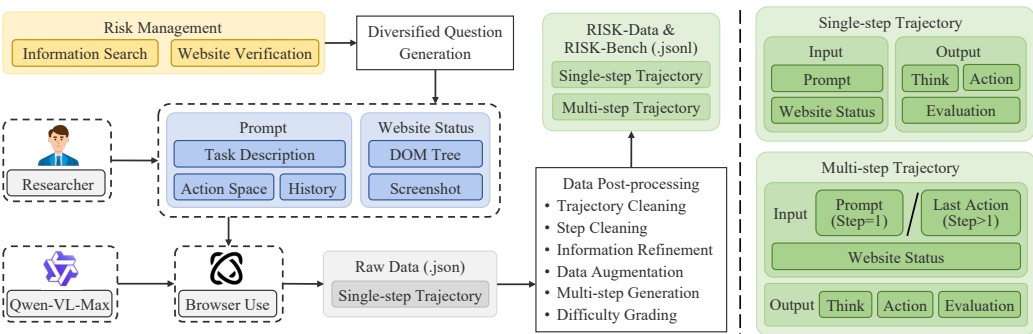

Figure 2: Data construction process for GUI agents in e-commerce risk management. By leveraging the Qwen-VL-Max, human-defined prompts, and diversified question templates, the Browser Use framework conducts multi-round interactions with webpages to collect raw data. Then, a series of data post-processing steps is applied to ensure the quality of the dataset.

## 3.2 DATA CONSTRUCTION

We develop a data construction pipeline, as illustrated in Figure 2, to gather high-quality data. The process begins with leveraging the capabilities of Qwen-VL-Max, a powerful vision-language model, to interact with webpages. We further design human-defined prompts and diversified question templates to guide the interactions, where domain-specific knowledge and scenarios are incorporated to ensure the relevance of the collected data. Based on these, the Browser Use framework facilitates multi-round interactions with various webpages, allowing for the collection of raw data that encompasses a wide range of scenarios encountered in e-commerce risk management.

Following data collection, we implement a series of post-processing steps to refine and curate the dataset. This includes (1) Trajectory Filtering: We filter out incomplete or unsuccessful interaction trajectories to ensure the meaningfulness of the data. (2) Step Cleaning: In the successful trajectories, there are some redundant or failed steps (e.g., repeatedly circumventing a slider captcha). We clean these steps to prevent the model from learning incorrect behaviors. (3) Information Refinement: We extract and structure the information (e.g., removing one-shot examples) from the raw data to facilitate easier access and analysis. (4) Data Augmentation: We apply various augmentation techniques to enhance the diversity and robustness of the dataset, such as paraphrasing questions and removing screenshots. (5) Multi-step generation: We generate multi-step interaction sequences by chaining together individual steps. As shown in Figure 2, we replace prompts with the last step's response after the first step, forming a "think-action-observation" loop and reducing trajectory length. Compared with single-step samples, this helps to simulate more complex scenarios that GUI agents may encounter in real-world applications. (6) Difficulty Grading: We categorize the data into different difficulty levels based on the accuracy of the advanced MLLM's response. This allows for a curriculum learning in the training process and a more nuanced evaluation of GUI agents' performance across varying levels of task complexity. Ultimately, we obtain a high-quality dataset and benchmark that effectively supports the development and assessment of GUI agents in this domain.

## 3.3 DATA STATISTICS

As shown in Appendix Table 5, RISK-Data comprises 8,492 single-step and 2,386 multi-step interaction trajectories, which are graded into three difficulty levels: easy, moderate, and difficult, based on the accuracy of the advanced MLLM's response [1]. In RISK-Data, the easy, moderate, and

---

[1] We use Qwen-VL-Max to answer each question 5 times. If the accuracy is 100%, 20-80%, and below 20%, we categorize the question as easy, moderate, and difficult, respectively.

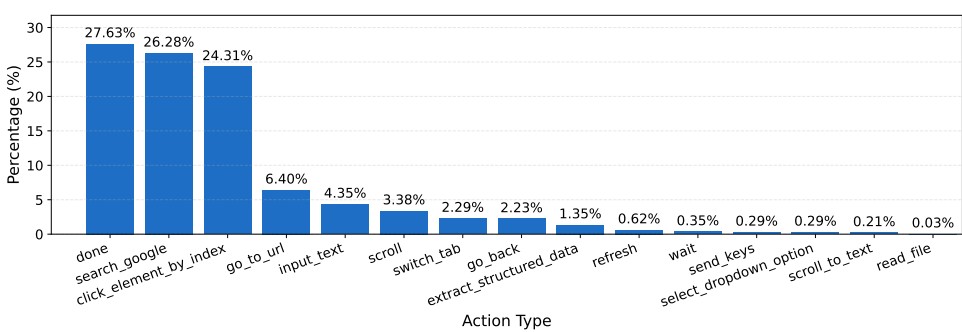

Figure 3: Action type distribution in RISK-Data, which includes 13 action types.

difficult samples account for 52%, 22%, and 26% in single-step tasks, and 36%, 14%, and 50% in multi-step tasks, respectively. Figure 3 elaborates the action type distribution in RISK-Data, which includes 13 action types. The most frequent action is `done`, accounting for 27.63%, followed by `search_google` (26.28%) and `click_element_by_index` (24.31%). We show action definitions in Appendix Table 6 and 7.

Appendix Figure 8 illustrates the token count and step count distribution of multi-step trajectories in RISK-Data, where we use the token count less than 21000 (around 82.94% of trajectories) for training because of the GPU memory limit. The minimum, maximum, and mean step count of trajectories are 4, 30, and 7.12, respectively. RISK-Bench consists of 802 single-step and 320 multi-step trajectories, where the easy, moderate, and difficult samples account for 47%, 25%, and 28% in single-step tasks, and 30%, 17%, and 53% in multi-step tasks, respectively. To ensure data integrity and prevent leakage, the samples in RISK-Bench are excluded from the training set.

## 4 METHODOLOGY

We propose an RFT framework based on GRPO, named RISK-R1, to train GUI agents. As shown in Figure 4, RISK-R1 consists of four key components in the reward function and policy optimization objective: (1) Updated format reward that enhances the syntactic correctness of the model's output and task understanding, (2) Stepwise accuracy reward that measures action accuracy considering both action completeness and training process, (3) Process reweight that emphasizes the step stage in the interaction process, and (4) Level reweight that focuses on different difficulty levels of tasks.

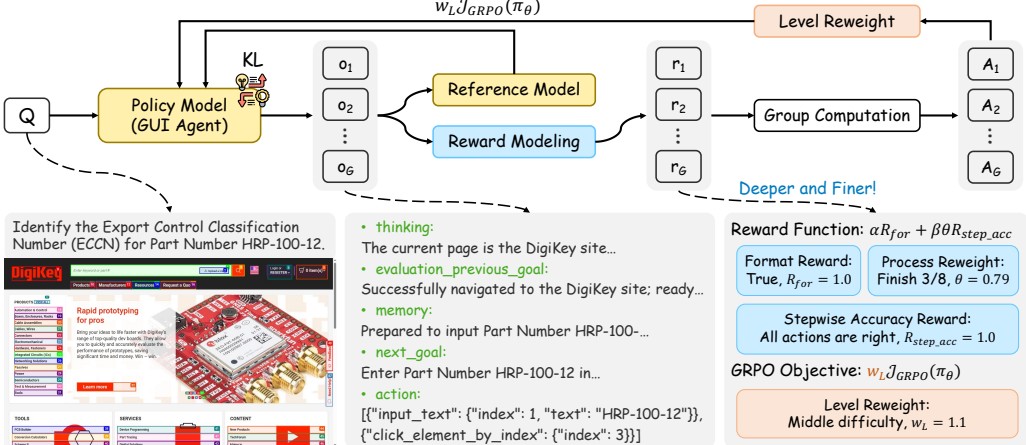

Figure 4: RISK-R1 framework. Our framework comprises four key components (format reward, stepwise accuracy reward, process reweight, and level reweight) in the reward function and policy optimization objective to effectively guide the learning process of GUI agents and enable a seamless transition from training to deployment.

### 4.1 PRELIMINARIES

The input at each step of a trajectory consists of the question $q \in Q$, the current webpage screenshot $I_t$, and the DOM tree $D_t$. The policy model takes these inputs and generates a set of candidate

responses $O = \{o_1, o_2, \ldots, o_G\}$. Each response $o_i$ will be evaluated by a reward model to obtain a reward score $r_i$. Then group computation is applied on these reward scores to estimate advantages: $A_i = \frac{r_i - mean(\{r_1, r_2, \cdots, r_G\})}{std(r_1, r_2, \cdots, r_G)}$. The policy model is then updated using the optimization objective:

$$\mathcal{J}_{\text{GRPO}}(\pi_\theta) = \mathbb{E}_{q \sim Q, \{o_i\}_{i=1}^G \sim \pi_{\theta_{\text{old}}}(\cdot|q)}$$

$$\frac{1}{G} \sum_{i=1}^G \frac{1}{|o_i|} \sum_{t=1}^{|o_i|} \left\{ \min\left[ \frac{\pi_\theta(o_{i,t}|q, o_{i,<t})}{\pi_{\theta_{\text{old}}}(o_{i,t}|q, o_{i,<t})} A_i, \text{clip}\left( \frac{\pi_\theta(o_{i,t}|q, o_{i,<t})}{\pi_{\theta_{\text{old}}}(o_{i,t}|q, o_{i,<t})}, 1-\epsilon, 1+\epsilon \right) A_i - \beta D_{\text{KL}}\left[ \pi_\theta \parallel \pi_{\text{ref}} \right] \right] \right\},$$

(1)

where $\epsilon$ controls the clipping range, $t$ is the token index in the response, and $\beta$ is the coefficient of the KL penalty to constrain the policy from deviating too far from the reference model $\pi_{\text{ref}}$.

## 4.2 Reward Design

In general GUI agents, the reward function typically focuses on the grounding accuracy of actions (e.g., the predicted point should be within the bounding box of the target element). However, this kind of reward function cannot satisfy the requirements of e-commerce risk management tasks due to complex webpages and diverse task scenarios. As shown in Figure 4, when there are dense elements on the page, the DOM tree structure is more suitable for accurate interaction. Therefore, a more dependable and comprehensive reward function is needed to guide the learning process of GUI agents in this domain. We detail our design below.

**Format Reward.** Format reward $R_{for}$ is introduced to ensure the syntactic and semantic correctness of the model's output. As 'think' content and 'action' content are still required in the output, we also consider the 'evaluation_previous_goal', 'memory', and 'next_goal' content, which come from the Browser Use framework and are beneficial for the model to understand the task process and webpage status. Among them, 'evaluation_previous_goal' is used to evaluate whether the last step's action is completed, 'memory' records the current task status, and 'next_goal' describes the next step's action. Moreover, since that RISK-R1 does not employ (x,y) coordinates for element selection but uses the element index in the DOM tree combined with the tools, we additionally check the correctness of the 'action' content format, which should be in the form of '[{<tool_name>:{<index>,<text>(optional)}}]'. This design ensures that the model's output is well-structured and interpretable, facilitating practical application in real scenarios. The format reward $R_{for}$ is 1 if the output format is correct, and 0 otherwise.

**Stepwise Accuracy Reward.** In practical tool calling scenarios, the tool list in the action predicted by the model may contain multiple actions to save the number of MLLM calls. Considering that the nature of RFT is to assist the model in exploring the correct path, the original binary accuracy reward treating the entire tool list as a whole is too coarse-grained to provide effective guidance at the early stage of training. Therefore, we propose a stepwise accuracy reward $R_{step\_acc}$ that evaluates the accuracy of each action in the list, providing more detailed feedback to the model at the early stage of training to facilitate exploration. After training the model to a certain extent, we further fine-tune it with the original binary accuracy reward to avoid the model exhibiting inertia under partial rewards. Specifically, for a tool list $T = \{t_1, t_2, \cdots, t_n\}$ in the action, where $t_i$ is the $i$-th tool in the list, we define the stepwise accuracy reward $R_{step\_acc}$ as follows:

$$R_{step\_acc} = \begin{cases} \frac{1}{n} \sum_{i=1}^n R_{acc}(t_i) & \text{early stage,} \\ R_{acc}(T) & \text{later stage,} \end{cases} \quad R_{acc}(t_i) = \begin{cases} 1 & \text{if } F_1(t_i, t_i^{gt}) > 0.5, \\ 0 & \text{otherwise,} \end{cases} \quad (2)$$

where $F_1(t_i, t_i^{gt})$ is the F1 score between the predicted tool $t_i$ and the ground truth tool $t_i^{gt}$.

**Process Reweight.** The motivation for process reweight comes from two aspects: (1) In business-oriented GUI tasks, the initial steps and associated webpage content are relatively simple (e.g., opening the Google search page), whereas later steps involve more complex pages (e.g., specific e-commerce pages), and (2) Early-stage steps exhibit high homogeneity, while later-stage steps show greater differentiation. Therefore, we propose a process reweighting $\theta$ to distinguish the importance of different steps in a trajectory, emphasizing the later steps that are more critical for task completion. We design a weight curve that increases with the step index, where the weight of the first step is $\gamma$ and the last step is 1.0, as shown in Appendix Figure 9. The process reweight $\theta$ for the $i$-th step in a trajectory with $n$ steps is defined as follows: $\theta(i) = \gamma + (1 - \gamma) \left( 1 + e^{-\left( 2\delta \frac{i-1}{n-1} - \delta \right)} \right)^{-1}$, where $\gamma$ and $\delta$ are hyperparameters to control the shape of the curve.

## 4.3 Reinforcement Learning Objective

To leverage the advantages of each component in the reward design, we combine them to form the overall reward $R$ for RISK-R1:

$$R = \alpha \cdot R_{for} + \beta \cdot \theta \cdot R_{step\_acc} \quad (3)$$

where $\alpha$ and $\beta$ are hyperparameters to balance the contributions of each component. Based on the overall reward $R$, we compute the advantage $A$ and optimize the policy model using the GRPO objective in Equation 1.

**Level Reweight.** In RISK-Data, the samples are graded into three difficulty levels: easy, moderate, and difficult. As demonstrated in (Zhou et al., 2025), question-level difficulty bias is beneficial for the model to focus on challenging aspects of the task. Therefore, we introduce a level reweight $w_{level}$ to adjust the contribution of samples at different difficulty levels to the objective function. Specifically, we set the level reweight $w_{level}$ for easy, moderate, and difficult samples as 1.0, 1.1, and 1.2, respectively. The final optimization objective of RISK-R1 is modified to $w_{level} \cdot \mathcal{J}_{GRPO}(\pi_\theta)$. Previously, the relative grounding box size was used to set the difficulty level, which is not appropriate since this criterion only considers the element size in isolation but

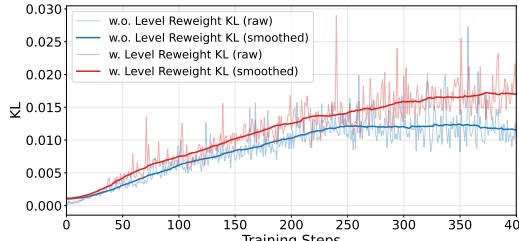

Figure 5: Comparison of KL divergence curves under different level reweight settings at early training stages, where the curve reflects the deviation of the policy from the reference model.

ignores the element density and page complexity. This limitation may lead to a wrong assessment of task difficulty, where empirical evidence is shown in Table 2. In contrast, our difficulty grading based on the advanced MLLM's accuracy is more comprehensive and dependable, where the comparison of KL divergence curves under different level reweight settings at early training stages is shown in Figure 5. It can be observed that the model with level reweighting deviates more and faster from the reference model, indicating that it explores more diverse strategies and learns more effectively from the challenging samples.

## 5 EXPERIMENTS

### 5.1 EXPERIMENTAL SETUP

**Implementation Details.** For SFT, we use the Qwen2.5-VL-7B-Instruct as the base model and train it for one epoch to learn the basic interaction capabilities. For RFT, we initialize the policy model with the supervised fine-tuned model and use the VeRL framework (Sheng et al., 2024) for training over six epochs. RFT Training is conducted on 8 NVIDIA H200-141G GPUs with the following hyperparameters: learning rate of 1e-6, rollouts per prompt of 8, and KL coefficient of 0.04. As the format has been initially standardized in SFT, we set reward coefficients $\alpha = 0.1$ and $\beta = 0.9$. The default process reweight coefficients are set to $\gamma = 0.7$ and $\delta = 4$. We use a stepwise reward in the first epoch and a binary reward in the remaining epochs.

**Training Datasets and Evaluation Benchmarks.** In SFT, we use all single-step and multi-step trajectories in RISK-Data for training. In RFT, we only use the single-step trajectories since the multi-step trajectories are too long to fit in the GPU memory. Considering general grounding data is beneficial for improving the model's website perception and element manipulation capabilities, we also incorporate the GUI-R1 (Luo et al., 2025) dataset into our training data. We evaluate RISK-R1 from three aspects: (1) Offline evaluation on RISK-Bench to assess the model's performance in e-commerce risk management tasks, (2) Offline evaluation on general GUI navigation benchmark OS-Genesis (Sun et al., 2024) to evaluate the model's generalization ability, where the web tasks are tested, and (3) Online evaluation in real-world e-commerce risk management scenarios to validate the practical effectiveness of RISK-R1.

We elaborate more experimental details in Appendix A.6.

### 5.2 MAIN RESULTS

**Offline Domain Evaluation.** We compare RISK-R1 with commercial models, general open-source models, and GUI-specific models on RISK-Bench and OS-Genesis, as shown in Table 1. RISK-R1-7B surpasses all baselines across single-step and multi-step tasks. Specifically, in single-step tasks on RISK-Bench, RISK-R1-7B attains an overall accuracy of 88.3%, outperforming GPT-4o by 6.8% and Qwen2.5-VL-72B by 7.7%. Notably, after RFT, RISK-R1-7B has a slight decrease of 0.3% in easy tasks but a substantial increase of 6.9% and 22.6% in moderate and difficult tasks, respectively. This change reveals that level reweighting effectively guides the model to focus on challenging samples, enhancing its problem-solving capabilities. In multi-step tasks, RISK-R1-7B achieves a task success rate of 82.8%, exceeding GPT-4o by 8.8%, indicating that multi-step trajectories in RISK-Data are beneficial for improving the model's task-level process understanding, while process reweighting emphasizes the importance of later steps in the trajectory, further enhancing performance.

**Offline General Evaluation.** In OS-Genesis evaluations, RISK-R1-7B attains a web task accuracy of 62.3%, surpassing GPT-4o by 7.0% and Qwen2.5-VL-72B by 12.3%. As web tasks in OS-Genesis also depend on the DOM tree structure, it shows superior capability by learning effective element selection strategies from RISK-

Table 1: Performance comparison of different models on RISK-Bench and OS-Genesis. The best results and the second best results are highlighted in **bold** and underline, respectively. Our RISK-R1-7B achieves SOTA performance, surpassing all baselines across single-step and multi-step tasks.

| Model | RISK-Bench | | | | | OS-Genesis |
| | Single-step | | | | Multi-step ↑ | Web Task ↑ |
| | Easy ↑ | Moderate ↑ | Difficult ↑ | Overall ↑ | | |
|---|---|---|---|---|---|---|
| *Commercial Models* | | | | | | |
| GPT-4o | 98.2 | 82.9 | 46.8 | 81.5 | 74.0 | 55.3 |
| Qwen-VL-Max | 95.8 | 78.5 | 22.4 | 72.9 | 50.0 | 50.3 |
| *General Open-source Models* | | | | | | |
| InternVL3-8B | 30.1 | 14.3 | 0.0 | 18.8 | 0.0 | 29.5 |
| Qwen2.5-VL-7B | 62.3 | 45.4 | 4.8 | 43.6 | 0.6 | 32.2 |
| Qwen2.5-VL-72B | 98.8 | 81.9 | 42.9 | 80.6 | 67.8 | 50.0 |
| *GUI-specific Models (SFT)* | | | | | | |
| UI-TARS-2B | 0.2 | 0.0 | 0.0 | 0.1 | 0.0 | 1.3 |
| UI-TARS-7B | 11.3 | 5.5 | 0.0 | 7.1 | 0.0 | 4.2 |
| UI-TARS-72B | 20.7 | 9.3 | 0.9 | 13.0 | 0.0 | 5.8 |
| OS-Atlas-7B | 37.3 | 27.7 | 2.4 | 26.1 | 0.0 | 23.0 |
| ShowUI-2B | 6.9 | 2.7 | 0.0 | 4.2 | 0.0 | 3.4 |
| Aguvis-7B | 8.3 | 4.5 | 0.0 | 5.3 | 0.0 | 29.7 |
| *GUI-specific Models (RL)* | | | | | | |
| GUI-R1-3B | 55.4 | 37.2 | 3.0 | 37.8 | 0.0 | 24.3 |
| GUI-R1-7B | 65.4 | 45.0 | 9.3 | 46.3 | 0.0 | 28.0 |
| InfiGUI-R1-3B | 18.6 | 11.7 | 1.4 | 12.6 | 0.0 | 10.6 |
| GUI-G1-3B | 55.1 | 38.9 | 4.5 | 38.4 | 0.0 | 19.1 |
| UI-TARS-1.5-7B | 44.9 | 28.4 | 1.9 | 30.1 | 0.0 | 26.5 |
| UI-Venus-Navi-7B | 0.7 | 0.0 | 0.0 | 0.3 | 0.0 | 14.3 |
| *Ours* | | | | | | |
| RISK-SFT-7B | **99.1** | 83.2 | 52.5 | 83.5 | 75.3 | 61.5 |
| RISK-R1-7B | 98.8 | **90.1** | **65.5** | **88.3** | **82.8** | **62.3** |

Table 2: Difficulty measurement analysis.

| Measurement | Single-step | Multi-step |
|---|---|---|
| No Reweighting | 86.7 | 79.6 |
| Rule Score | 86.1 (-0.6) | 78.0 (-1.6) |
| LLM Response | 88.3 (+1.6) | 82.8 (+4.8) |

Table 3: Difficulty weight configurations.

| Configuration | Single-step | Multi-step |
|---|---|---|
| {0.8,0.9,1.0} | 87.8 | 82.0 |
| {1.0,1.1,1.2} | 88.3 | 82.8 |
| {1.0,1.3,1.5} | 88.1 | 82.4 |

Data. These results demonstrate the effectiveness of the RISK-R1 framework in enhancing the capabilities of GUI agents for e-commerce risk management, while not compromising their generalization ability.

**Online Evaluation.** For read-time multi-step decision-making evaluation, we use the Browser-Use framework to build a webpage interaction environment and compare RISK-R1 with various baselines. Different from offline evaluations, the model may encounter unseen webpages or changed page structures (even if the objective website is the same) during online evaluations, which poses a greater challenge to the model's generalization ability and robustness. As shown in Table 4, although the task completion rate of RISK-R1-7B is slightly lower than that of Qwen2.5-VL-72B, it achieves the highest task success rate of 70.5%, outperforming Qwen2.5-VL-72B by 1.6% and Qwen-VL-Max by 4.3%. This indicates that RISK-R1-7B can effectively complete tasks even in complex and dynamic real-world scenarios, demonstrating its practical applicability in e-commerce risk management.

## 5.3 REWARD DESIGN ANALYSIS

**Level Reweight: Difficulty Grading Matters.** In RISK-R1, we use the advanced MLLM's accuracy to grade the difficulty of samples and set the level reweight accordingly. To validate the effectiveness of this grading method, we compare it with no reweighting and rule-based scoring methods, as shown in Table 2. The rule-based scoring method assigns difficulty levels based on the tool count at each step, where easy, moderate, and difficult samples contain 1, 2, and more than 2 tools, respectively. The results indicate that inappropriate difficulty grading methods can negatively impact model performance, with the rule-based scoring method leading

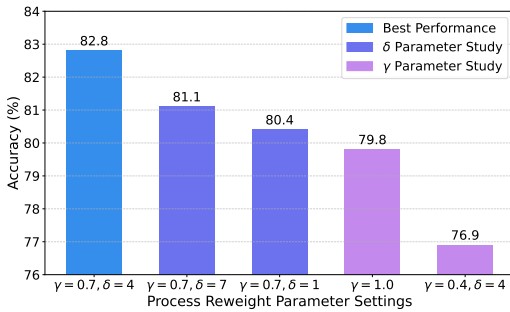
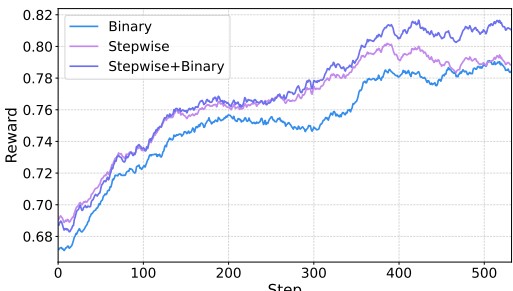

Figure 6: Hyperparameter sensitivity analysis for process reweighting. The best performance is achieved at $\gamma = 0.7$ and $\delta = 4$.

Figure 7: Stepwise accuracy reward analysis. Peak performance is achieved at the combination of stepwise and binary accuracy rewards.

to decreases of 0.6% and 1.6% in single-step and multi-step tasks, respectively. Another notable setting is the difficulty weight configuration, which influences the emphasis on challenging samples. As shown in Table 3, the configuration {1.0, 1.1, 1.2} yields the best performance, while both overly flat ({0.8, 0.9, 1.0}) and overly steep ({1.0, 1.3, 1.5}) configurations lead to performance drops. This analysis highlights the importance of appropriate difficulty grading and weight configuration in guiding the reinforcement learning optimization.

**Process Reweight: Critical Steps in Task Completion.** The goal of process reweighting is to emphasize the importance of later steps in a trajectory, which are more distinguishable and determine the success of task completion. We conduct hyperparameter sensitivity analysis for process reweight, as shown in Figure 6. $\gamma$ controls the weight of the initial step and $\delta$ controls the growth rate of the weight curve, where the visualization is shown in Figure 9. Comparison results reveal that an appropriate setting of process reweight can guide the model to focus on critical steps. However, an excessively low $\gamma$ (e.g., 0.4) or an excessively high $\delta$ (e.g., 7) leads to performance degradation, as the model may neglect the importance of early and intermediate steps, leading to suboptimal learning outcomes.

**Stepwise Accuracy Reward: Fine-grained Feedback for Exploration.** RISK-R1 employs stepwise accuracy reward in the early stage of training to provide more fine-grained feedback for the model, facilitating exploration. We analyze the impact of different reward settings, as shown in Figure 7 (All reward curves are smoothed). Leveraging stepwise accuracy reward in the early stage provides a faster reward enhancement by encouraging the model to learn from partially correct tool calls. Nevertheless, using stepwise accuracy reward throughout the entire training process does not yield better results than the solely binary accuracy reward, as the model may develop inertia under partial rewards. The optimal approach is to combine both reward types, using stepwise accuracy reward in the early stage and binary accuracy reward in the later stage, which effectively balances exploration and exploitation.

Table 4: Performance comparison of models on RISK-Bench with online evaluation, where webpage interaction is built on Browser-Use. Compared with SOTA baselines, RISK-R1-7B achieves the highest task success rate while maintaining a competitive task completion rate.

| Model | Completion | Success |
|---|---|---|
| *Commercial Models* | | |
| Qwen-VL-Max | 85.2 | 66.2 |
| *General Open-source Models* | | |
| InternVL3-8B | 0.0 | 0.0 |
| Qwen2.5-VL-7B | 8.3 | 46.4 |
| Qwen2.5-VL-72B | **88.7** | 68.9 |
| *GUI-specific Models (SFT)* | | |
| UI-TARS-7B | 0.0 | 0.0 |
| UI-TARS-72B | 0.0 | 0.0 |
| OS-Atlas-7B | 4.4 | 37.2 |
| ShowUI-2B | 0.0 | 0.0 |
| *GUI-specific Models (RL)* | | |
| GUI-R1-7B | 0.0 | 0.0 |
| InfiGUI-R1-3B | 0.0 | 0.0 |
| GUI-G1-3B | 0.0 | 0.0 |
| UI-TARS-1.5-7B | 0.0 | 0.0 |
| *Ours* | | |
| RISK-SFT-7B | 86.1 | 67.0 |
| RISK-R1-7B | 87.6 | **70.5** |

## 6    Conclusion

In this work, we aim to address the critical challenge of automating e-commerce risk management tasks that involve dynamic, multi-step web interactions. Specifically, we propose the RISK framework, which incorporates a domain-specific dataset RISK-Data, a benchmark RISK-Bench, and a novel RFT approach RISK-R1 that comprises a comprehensive reward function and optimization objective to guide the model's learning process. The experimental results confirm that RISK-R1 outperforms existing methods, showing a 6.8% improvement in single-step and an 8.8% improvement in multi-step performance, as well as achieving a top task success rate of 70.5% in real-world web environments. Our work provides a scalable, domain-specific solution for automating complex web interactions in high-stakes compliance and risk management tasks.

## 7 ETHICS STATEMENT

In this work, we ensure ethical compliance by sourcing all data exclusively from publicly available websites, with no personally identifiable information (PII) or sensitive data included. Strict data anonymization protocols are implemented to safeguard user privacy and address potential concerns.

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

## A  APPENDIX

### A.1  LIMITATIONS AND FUTURE WORK

**Limitations.** Although RISK-R1 demonstrates superior performance in e-commerce risk management tasks, there are still some limitations. (1) The current multi-step trajectories in RISK-Data are mainly used in SFT, while RFT only utilizes single-step trajectories due to GPU memory constraints. This may limit the model's ability to fully learn multi-step decision-making processes. (2) Although we have incorporated process reweighting in our RFT framework to simulate an offline multi-step webpage interaction process, it may not fully capture the complexities and diversity of real-world scenarios. An online reinforcement learning framework could be more effective in this regard.

**Future Work.** Given the limitations mentioned above, we plan to address them in future work. We note that Shi et al. (2025) proposes a mobile GUI agent framework that leverages reinforcement learning in an online environment. We intend to adapt this framework to web-based GUI agents, enabling the model to learn directly from real-time interactions. Through this approach, GPU memory constraints can be alleviated and the model's multi-step decision-making capabilities can be further enhanced. Additionally, we plan to build a high-concurrency cluster of browser environments to collect more diverse and complex multi-step instances, further enriching RISK-Data.

### A.2  TASK DEFINITION

E-commerce risk management mainly involves two aspects: (1) Information Search: external information retrieval and extraction for risk intelligence, and (2) Website Verification: website authenticity verification for risk intelligence. The specific tasks are described as follows:

#### A.2.1  EXTERNAL INFORMATION RETRIEVAL AND EXTRACTION FOR RISK INTELLIGENCE

This module is designed to autonomously interact with external websites, including search engines, e-commerce platforms, enterprise registries, logistics trackers, and customs clearance portals. The collected information supports multi-dimensional tasks such as risk profiling, fraud detection, anti-money laundering(AML) compliance, and regulatory verification.

**Product Risk Profile.** To satisfy regulatory compliance and risk management requirements, it is essential to incorporate external data sources in constructing the product risk profiles associated with a given transaction. Such profiles encompass product-specific risk attributes, including legal and regulatory restrictions, HS code, pricing irregularities, and other indicators pertinent to trade-based risk assessment.

**Merchant Risk Profile.** Acquiring legal registration details, business licenses, ownership and control structures, scope of operations, certifications, and related entities to assess beneficial ownership and detect shell companies or high-risk partnerships.

**Client Risk Profile.** Collecting publicly available identifiers such as registered emails, phone numbers, and cross-referenced identity records to assist in customer verification, fraud prevention, and AML compliance.

**Logistics and Supply Chain Tracking.** Monitoring shipping status (dispatch, in transit, customs clearance, final delivery) through courier, freight, or e-commerce logistics platforms, supporting trade verification and trade model restoration.

**Customs Declaration & Clearance Status Audit.** Accessing customs or import/export systems to verify declaration completion, inspection results, release status, and anomalies that may indicate misdeclaration or sanctions evasion.

#### A.2.2  WEBSITE AUTHENTICITY VERIFICATION FOR RISK INTELLIGENCE

The module is designed to automate the validation of the legitimacy, security, and regulatory compliance of websites, merchant portals, and transaction endpoints, thereby mitigating phishing, spoofing, and fraudulent transaction risks.

**Transaction Laundry Detection.** Identifying unauthorized or illicit content embedded under legitimate merchant domains, including gambling, adult services, fraudulent financial offerings, and money laundering transaction pathways.

**Website Accessibility and Identity Verification.** Assessing reachability (HTTP status codes, response latency), SSL/TLS certificate validity, and WHOIS/domain registration congruence with officially filed corporate identities—reducing exposure to impersonation threats.

**Content Consistency Assurance.** Cross-verifying brand, product, and company registration data across multiple site sections or historical versions to prevent brand hijacking, data manipulation, or asymmetric disclosures used in fraud scenarios.

**Secure Payment Channel Validation.** Verifying the legitimacy of payment processors, detecting high-risk payment mechanisms (anonymous crypto transfers, non-compliant third-party gateways), and ensuring domain consistency between payment pages and main sites to prevent phishing and mitigate fund diversion risks.

### A.3 Multi-step Trajectory Statistics

We provide the statistics of RISK-Data and RISK-Bench in Table 5. The token count distribution and step count distribution of multi-step trajectories are shown in Figure 8.

Table 5: Statistics of RISK-Data and RISK-Bench. Note that RISK-Bench is additionally collected for evaluation, and this part of data is not used during training for data leakage prevention.

| Data | Trajectory | Size | Test Capability | Grading |
|------|-----------|------|-----------------|---------|
| RISK-Data | Single-step | 8,492 | Accuracy of Webpage Perception and Element Manipulation | Easy: 52%, Moderate: 22%, Difficult: 26% |
| | Multi-step | 2,386 | Task-level process understanding, planning, and correction capability | Easy: 36%, Moderate: 14%, Difficult: 50% |
| RISK-Bench | Single-step | 802 | Accuracy of Webpage Perception and Element Manipulation | Easy: 47%, Moderate: 25%, Difficult: 28% |
| | Multi-step | 320 | Task-level process understanding, planning, and correction capability | Easy: 30%, Moderate: 17%, Difficult: 53% |

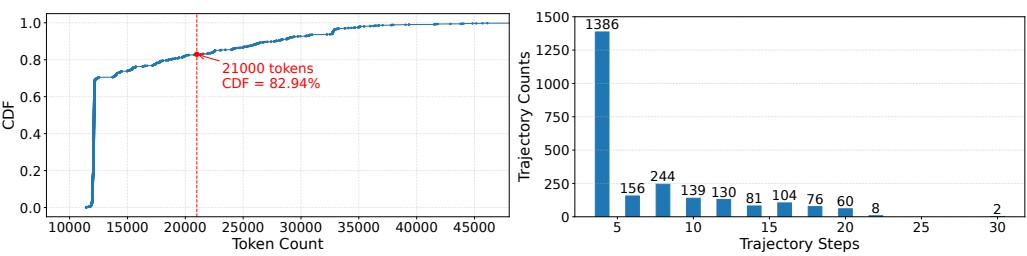

(a) Multi-step trajectory token count distribution    (b) Multi-step trajectory step count distribution

Figure 8: Token count distribution and step count distribution of multi-step trajectories, where we use the token count of trajectories less than 21000 for training because of the GPU memory limit. The minimum, maximum, and mean step count of trajectories are 4, 30, and 7.12, respectively.

### A.4 Action Definition

There are 13 actions in total used in RISK, and their definitions are shown in Table 6 and Table 7.

### A.5 Visualization of Weight Curve for Process Reweight

Visualization of weight curve for process reweight is shown in Figure 9.

### A.6 Experimental Details

**Implementation Details.** For SFT, we use the Qwen2.5-VL-7B-Instruct as the base model and train it for one epoch to learn the basic interaction capabilities. For RFT, we initialize the policy model with the supervised fine-tuned model and use the VeRL framework (Sheng et al., 2024) for training over six epochs. RFT Training is conducted on 8 NVIDIA H200-141G GPUs with the following hyperparameters: learning rate of 1e-6, rollouts per prompt of 8, and KL coefficient of 0.04. As the format has been initially standardized in SFT, we set reward coefficients $\alpha = 0.1$ and $\beta = 0.9$. The default process reweight coefficients are set to $\gamma = 0.7$ and $\delta = 4$.

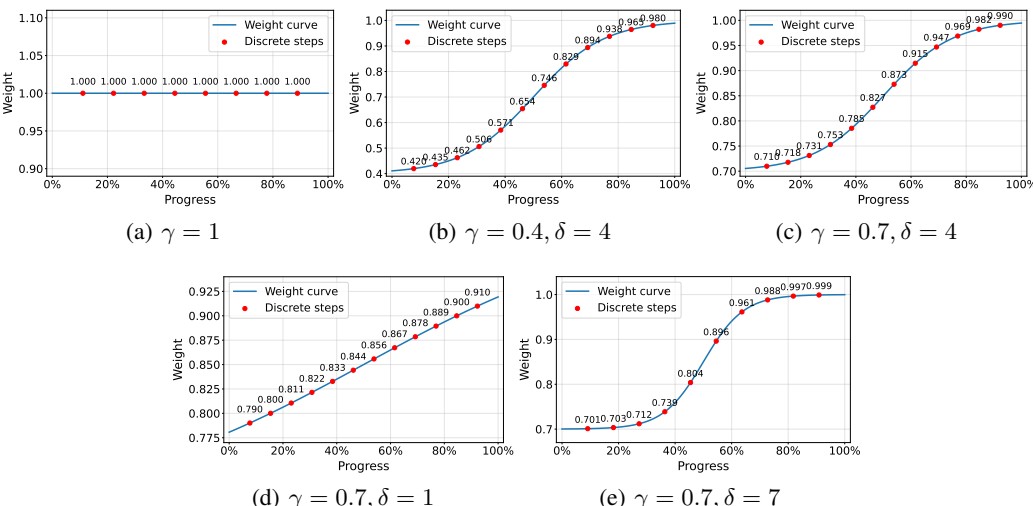

Figure 9: Weight curve for process reweighting.

We use a stepwise reward in the first epoch and a binary reward in the remaining epochs. During inference, we deploy the vLLM engine (Kwon et al., 2023) with a temperature of 0 to generate deterministic responses.

**Training Datasets and Evaluation Benchmarks.** In SFT, we use all single-step and multi-step trajectories in RISK-Data for training, where the maximum of image pixels and token length are set to 1176000 and 21000, respectively. Trajectories with token length exceeding 21000 are excluded rather than truncated to avoid incomplete information. In RFT, we only use the single-step trajectories since the multi-step trajectories are too long to fit in the GPU memory. We set the maximum image pixels to 1176000 and the maximum token length to 13824. Considering general grounding data is beneficial for improving the model's website perception and element manipulation capabilities, we also incorporate 3570 grounding samples from the GUI-R1 dataset into our training data. We evaluate RISK-R1 from three aspects: (1) Offline evaluation on RISK-Bench to assess the model's performance in e-commerce risk management tasks, (2) Offline evaluation on general GUI navigation benchmark OS-Genesis (Sun et al., 2024) to evaluate the model's generalization ability, where the web tasks are tested, and (3) Online evaluation in real-world e-commerce risk management scenarios to validate the practical effectiveness of RISK-R1. All experimental results of baselines are obtained by re-testing with the same prompts and tools as RISK-R1 for fair comparison.

**Evaluation Metrics.** In offline single-step trajectory evaluations, we use the accuracy of tool calls as the evaluation metric, where a tool call is considered correct if its F1 score with the ground truth tool call exceeds 0.5. In offline multi-step trajectory evaluations, we use the task success rate as the evaluation metric, where a trajectory is considered successfully completed if all tool calls in the trajectory are correct. In online evaluations, we use the task completion rate and task success rate as the evaluation metrics, where the task completion rate is the percentage of tasks completed within a limited number of steps (set to 20), and the task success rate is the percentage of tasks successfully completed.

## A.7 ABLATION STUDY

**Coefficients of Reward Components.** We conduct ablation studies on the coefficients of reward components, as shown in Table 8. The results indicate that both format reward and stepwise accuracy reward are essential for RISK-R1, as removing format reward ($\alpha = 0.0$) or reducing the weight of stepwise accuracy reward ($\beta = 0.5$) leads to performance degradation. The optimal configuration is $\alpha = 0.1$ and $\beta = 0.9$, which balances the contributions of each component.

Table 8: Proportion of Difficulty.

| $\alpha$ | $\beta$ | Single-step | Multi-step |
|------|------|-------------|------------|
| 0.5 | 0.5 | 86.7 | 81.9 |
| 0.1 | 0.9 | 88.3 | 82.8 |
| 0.0 | 1.0 | 86.5 | 80.3 |

Table 6: All actions and their definitions used in RISK (Part 1).

| Action | Definition |
|---|---|
| `search_google:  {'query':  {'type':  'string'}}` | Search the query in Google. The query should be a search query like human search in Google, concrete and not vague or super long. |
| `done:  {'text':  {'type': 'string'}, 'success':  {'type': 'boolean'}, 'files_to_display': {'anyOf':  [{'items':  {'type': 'string'}, 'type':  'array'}, {'type':  'null'}], 'default': []}}` | Complete task - provide a summary of results for the user. Set `success=True` if task completed successfully, false otherwise. Text should be your response to the user summarizing results. Include files you would like to display to the user in `files_to_display`. |
| `click_element_by_index: {'index':  {'type':  'integer'}, 'delay':  {'anyOf':  [{'type': 'integer'}, {'type':  'null'}], 'default':  None, 'description': 'Time to wait between `mousedown` and `mouseup` in milliseconds.  Defaults to 0.'}}` | Click element by index. If needed, use delay for mouse hold. |
| `scroll:  {'down':  {'type': 'boolean'}, 'num_pages': {'type':  'number'}, 'index':  {'anyOf':  [{'type': 'integer'}, {'type':  'null'}], 'default':  None}}` | Scroll the page by specified number of pages (set `down=True` to scroll down, `down=False` to scroll up, `num_pages=number` of pages to scroll like 0.5 for half page, 1.0 for one page, etc.). Optional index parameter to scroll within a specific element or its scroll container (works well for dropdowns and custom UI components). |
| `switch_tab:  {'page_id': {'type':  'integer'}}` | Switch to a different tab. |
| `go_back:  {}` | Go back to the previous page. |
| `extract_structured_data: {'query':  {'type':  'string'}, 'extract_links':  {'type': 'boolean'}}` | Extract structured, semantic data (e.g. product description, price, all information about XYZ) from the current webpage based on a textual query. This tool takes the entire markdown of the page and extracts the query from it. Set `extract_links=True` ONLY if your query requires extracting links/URLs from the page. Only use this for specific queries for information retrieval from the page. Don't use this to get interactive elements - the tool does not see HTML elements, only the markdown. |
| `input_text:  {'index':  {'type': 'integer'}, 'text':  {'type': 'string'}}` | Click and input text into a input interactive element. |
| `refresh:  {}` | Refresh the current page. |
| `wait:  {'seconds':  {'default': 3, 'type':  'integer'}}` | Wait for a specified duration (default 3 seconds). |
| `scroll_to_text:  {'text': {'type':  'string'}}` | Scroll to the specified text in the current page. |
| `go_to_url:  {'url':  {'type': 'string'}, 'new_tab':  {'type': 'boolean'}}` | Navigate to URL, set `new_tab=True` to open in new tab, False to navigate in current tab. |

Table 7: All actions and their definitions used in RISK (Part 2).

| Action | Definition |
|---|---|
| `read_file: {'file_name': {'type': 'string'}}` | Read `file_name` from file system. |
| `send_keys: {'keys': {'type': 'string'}}` | Send strings of special keys to use Playwright page.keyboard.press - examples include Escape, Backspace, Insert, PageDown, Delete, Enter, or Shortcuts such as 'Control+o', 'Control+Shift+T'. |
| `select_dropdown_option: {'index': {'type': 'integer'}, 'text': {'type': 'string'}}` | Select dropdown option for interactive element index by the text of the option you want to select. |

