# OpenReview forum: "RISK: A Framework for GUI Agents in E-commerce Risk Management"
_ICLR.cc/2026/Conference — ICLR 2026 Conference Withdrawn Submission_

### Official Review · Reviewer_85T4 · 2025-10-31

**Soundness:** 2
**Presentation:** 2
**Contribution:** 1
**Rating:** 2
**Confidence:** 3

**Summary:**

Decision making in e-commerce settings requires aggregation of diverse data, which is challenging. GUI-agents in literature are mostly limited to single-step operations, which renders them ineffective for e-commerce settings. Also, because of domain-specific dataset, it is not straightforward to develop GUIs specifically for e-commerce applications. This paper collects a dataset for e-commerce applications using a VLM (Qwen-VL-Max) and curates the data, which contains diverse information for such applications. In addition, a subset of the data is employed as a benchmark for evaluating GUI agents in e-commerce settings. Lastly, the paper applies specific reward designs over GRPO to enable GUI agents work well in e-commerce settings. Overall, proposed approach performs well in e-commerce risk management tasks.

**Strengths:**

- **Dataset**: The created dataset and the benchmark would be useful for e-commerce community.
- **Performance**: The proposed approach shows performance improvements over baselines in e-commerce tasks.

**Weaknesses:**

- **Effectiveness**: The paper's primary claim to effectiveness (Table 1) seems to be based on an unfair comparison. It appears the proposed dataset was only used to train the RISK-SFT-7B and RISK-R1-7B models, while the baselines were not fine-tuned on this data. If this is true, the comparison is invalid, as the improvements may stem entirely from the new training data, not the proposed method. Please clarify this and provide a fair, head-to-head comparison where all models are trained on the same data.

- **Novelty**:  The paper's main methodological contribution appears to be the specific reward designs for GRPO. However, this fails to discuss or compare against a large body of prior work on similar ideas, such as stepwise rewards, process-level optimization, and token-level preference modeling [a-f]. The paper must clearly differentiate its contribution from this existing work, as the methodological novelty is currently questionable beyond the experimental setting.

[a]: Lai, Xin, et al. "Step-dpo: Step-wise preference optimization for long-chain reasoning of llms." arXiv preprint arXiv:2406.18629 (2024).

[b]: Wu, Junkang, et al. "$\beta $-DPO: Direct Preference Optimization with Dynamic $\beta$." Advances in Neural Information Processing Systems 37 (2024): 129944-129966.

[c]: Zeng, Yongcheng, et al. "Token-level Direct Preference Optimization." International Conference on Machine Learning. PMLR, 2024.

[d]: Zhang, Jixiao, and Chunsheng Zuo. "Grpo-lead: A difficulty-aware reinforcement learning approach for concise mathematical reasoning in language models." arXiv preprint arXiv:2504.09696 (2025).

[e]: Xiong, Weimin, et al. "Watch Every Step! LLM Agent Learning via Iterative Step-level Process Refinement." Proceedings of the 2024 Conference on Empirical Methods in Natural Language Processing. 2024.

[f]: Lightman, Hunter, et al. "Let's verify step by step." The Twelfth International Conference on Learning Representations. 2023.

- **Missing Details of Dataset Creation**: The created dataset is the core contribution of this paper. However, the data construction pipeline in 3.2 is vague. Critical details are missing, such as the exact "human-defined prompts" and "question templates" used, which makes the dataset collection process irreproducible.

- **Ablation of Reward Components**: Paper applies specific reward designs over GRPO. However, reward components are not ablated. It is impossible to know which parts of the reward design (if any) are responsible for the claimed performance gains.

**Questions:**

- How you created the dataset collection pipeline? How you combined Qwen-VL-Max with Browser Use?
- What are human-defined prompts, and which question templates are used for dataset collection? Also, how did you incorporate domain-specific knowledge into dataset collection pipeline?
- How each reward component affects the final-performance of the model?
- In table 3, three different level re-weight sets are analyzed. Can you please analyze broader settings, such as steeper configurations?

---

### Official Review · Reviewer_vRKd · 2025-11-01

**Soundness:** 2
**Presentation:** 3
**Contribution:** 2
**Rating:** 4
**Confidence:** 4

**Summary:**

This paper introduces RISK, a framework addressing the automation of e-commerce risk management tasks, which demand aggregating deeply embedded web data via multi-step, stateful GUI interactions. Existing agents often fail this requirement. RISK integrates a domain-specific dataset (RISK-Data, 8,492 single-step and 2,386 multi-step trajectories), a graded benchmark (RISK-Bench), and RISK-R1, a tailored reinforcement fine-tuning (RFT) approach. The method uses specialized rewards (format, stepwise accuracy, process/level reweighting) to guide learning, demonstrating improved offline single-step (6.8%) and multi-step (8.8%) performance over baselines.

**Strengths:**

The primary contribution is the development of a domain-specific solution for high-stakes e-commerce risk management, specifically targeting the limitations of existing GUI agents in handling complex, dynamic multi-step tasks. The construction of RISK-Bench and RISK-Data provides necessary resources for standardized evaluation in this specialized area. Furthermore, the novel RFT reward structure in RISK-R1, including process reweight and the advanced MLLM-based level reweighting, effectively guides the model to focus on challenging samples and later, critical steps in the sequence. However, the achieved SOTA result (70.5% online success) must be viewed cautiously, as the RFT phase primarily relied on single-step trajectories due to documented GPU memory constraints.

**Weaknesses:**

This paper is about a vertical domain dataset and model. However, for such a paper, we expect to see: 1. The highly distinct characteristics of the data and evaluation methods within the domain. 2. How the model's architecture and insights differ from those in general domains. 3. The demonstration of the domain's value. 4. Comprehensive open-sourcing. I believe the paper's discussion is lacking on every one of these points, and therefore, it is not an ICLR-level paper.

**Questions:**

1. How are the essential characteristics of the "risk management" business domain (e.g., information reliability, time sensitivity, subtlety of fraud) reflected in the data structure and evaluation metrics?
2. How can you demonstrate that these new reward mechanisms are designed specifically for the unique challenges of the e-commerce risk management domain, rather than being merely general RFT optimization?
3. How do you justify the real-world "value" in this vertical scenario using the technical "success rate"? What are your thoughts on this?
4. Given the dynamic nature of web content, how do you ensure the reusability and maintainability of the dataset? What are the open-source plans?

---

### Official Review · Reviewer_we6s · 2025-11-02

**Soundness:** 3
**Presentation:** 3
**Contribution:** 2
**Rating:** 4
**Confidence:** 3

**Summary:**

The paper presents a comprehensive system for building and training graphical user interface (GUI) agents to automate complex e-commerce risk management workflows. It proposes an integrated framework combining a curated dataset (RISK-Data), an evaluation benchmark (RISK-Bench), and a reinforcement fine-tuning method (RISK-R1). The goal is to enable agents capable of handling multi-step, stateful interactions across dynamic webpages, a challenge that existing GUI models typically fail to address.

**Strengths:**

1.The construction of RISK-Data is executed through a rigorous human-in-the-loop pipeline with multi-step refinement.
2. The paper maintains strong reproducibility by providing implementation details, GPU setups.
3. It proposes a novel multi-component reward design

**Weaknesses:**

1. Although the paper motivates e-commerce risk management, broader generalization to other compliance domains (finance, healthcare) is not demonstrated.
2. The paper heavily focuses on quantitative metrics. Case studies or visualizations of GUI interactions (e.g., step reasoning chains) would help interpret how RISK-R1 improves over baselines.
3. Since RISK-Data and RISK-Bench are curated by the authors using Browser-Use and Qwen models, there is potential overlap or bias toward model-friendly environments.

**Questions:**

1. Would RISK-R1 generalize to larger models?
2. What is the fundamental differences between RISK-bench and previous agentic benchmarks? If differences are few, is there any stronger baselines should be included?

---

### Note · Authors · 2025-12-22

I have read and agree with the venue's withdrawal policy on behalf of myself and my co-authors.